# New Insights on the PBMCs Phospholipidome in Obesity Demonstrate Modulations Associated with Insulin Resistance and Glycemic Status

**DOI:** 10.3390/nu13103461

**Published:** 2021-09-29

**Authors:** Chloé Wilkin, Megan Colonval, Jonas Dehairs, Nathalie Esser, Margaud Iovino, Marco A. Gianfrancesco, Marjorie Fadeur, Johan V. Swinnen, Nicolas Paquot, Jacques Piette, Sylvie Legrand-Poels

**Affiliations:** 1Laboratory of Immunometabolism and Nutrition, GIGA-Inflammation, Infection & Immunity, University of Liège, 4000 Liège, Belgium; c.wilkin@uliege.be (C.W.); mcolonval@outlook.com (M.C.); nathalie.esser@chuliege.be (N.E.); margaud.iovino@doct.uliege.be (M.I.); marco.gianfrancesco@icloud.com (M.A.G.); marjorie.fadeur@chuliege.be (M.F.); nicolas.paquot@chuliege.be (N.P.); 2Laboratory of Lipid Metabolism and Cancer, Department of Oncology, KU Leuven, 3000 Leuven, Belgium; jonas.dehairs@kuleuven.be (J.D.); j.swinnen@kuleuven.be (J.V.S.); 3Division of Diabetes, Nutrition and Metabolic Disorders, Department of Medicine, University Hospital of Liège, 4000 Liège, Belgium; 4Laboratory of Virology and Immunology, GIGA-Molecular Biology of Diseases, University of Liège, 4000 Liège, Belgium; jpiette@uliege.be

**Keywords:** lipidomic, phospholipids, membrane lipids, metabolism, obesity, type 2 diabetes, immunology

## Abstract

(1) Background: Obesity and type 2 diabetes have been suspected to impact both intrinsic metabolism and function of circulating immune cells. (2) Methods: To further investigate this immunometabolic modulation, we profiled the phospholipidome of the peripheral blood mononuclear cells (PBMCs) in lean, normoglycemic obese (OBNG) and obese with dysglycemia (OBDysG) individuals. (3) Results: The global PBMCs phospholipidome is significantly downmodulated in OBDysG unlike OBNG patients when compared to lean ones. Multiple linear regression analyses show a strong negative relationship between the global PBMCs phospholipidome and parameters assessing insulin resistance. Even though all classes of phospholipid are affected, the relative abundance of each class is maintained with the exception of Lyso-PC/PC and Lyso-PE/PE ratios that are downmodulated in PBMCs of OBDysG compared to OBNG individuals. Interestingly, the percentage of saturated PC is positively associated with glycated hemoglobin (HbA1c). Moreover, a few lipid species are significantly downmodulated in PBMCs of OBDysG compared to OBNG individuals, making possible to distinguish the two phenotypes. (4) Conclusions: This lipidomic study highlights for the first-time modulations of the PBMCs phospholipidome in obese patients with prediabetes and type 2 diabetes. Such phospholipidome remodeling could disrupt the cell membranes and the lipid mediator’s levels, driving an immune cell dysfunction.

## 1. Introduction

Obesity is characterized by a chronic low-grade inflammation that is initiated in adipose tissue (AT) and interferes with insulin signaling and ultimately contributes to type 2 diabetes (T2D) [1,2]. Beside the activation of immune cells contributing to this sustained inflammation, obesity has also detrimental effects on immunity as evidenced by higher rates of vaccine failure, complications from infection [3,4,5,6] and increased risk of cancer [7,8,9]. These negative effects include the disruption of lymphoid tissue integrity, alterations in the development, phenotype and activity of leukocytes as well as an impairment of the crosstalk between innate and adaptive immune responses [3].

Moreover, it has recently emerged that systemic metabolic disorders associated with obesity may in turn impact both intrinsic metabolism and function of circulating immune cells. Indeed, the activation, growth, proliferation, engagement of effector functions, and return to homeostasis of immune cells are intimately linked with dynamic changes in cellular metabolism [10]. Therefore, a microenvironment enriched or depleted in some metabolites can disrupt both intrinsic metabolism and activation of immune cells [11]. This « immunometabolic » concept is very well illustrated in the case of peripheral Natural Killer (NK) cells in obesity [12]. Authors demonstrated that increased levels of circulating free fatty acids (FFAs) in obesity induce triglycerides (TG) synthesis in peripheral NK cells. TG accumulation in lipid droplets, in turn, leads to the glucose metabolism paralysis associated with an impairment of the NK cells cytotoxicity and a loss of tumor immunosurveillance [12].

As modifications in plasma FFAs and lipids levels, an increase in blood glucose concentration could have an impact on both intrinsic metabolism and functions of peripheral blood mononuclear cells (PBMCs). It’s known that the rate of glucose utilization can be modulated by its availability [13]. Thus, a hyperglycemic environment could exacerbate the glucose metabolism and glycolysis-dependent functions in circulating immune cells. For example, the upregulation of the pro-inflammatory glycolysis-dependent properties of trained atherogenic macrophages was proposed to mediate the higher susceptibility of diabetic patients to develop cardiovascular diseases [14].

Besides glycolysis, other metabolic pathways such as those involving lipids or amino acids can also instruct the functions of immune cells [10]. For example, the TG synthesis was recently shown to play a key role in the pro-inflammatory activation of ex vivo macrophages [15]. What about the metabolism of glycerophospholipids and sphingolipids in circulating immune cells of obese and T2D patients? The glycerophospholipids are the building blocks of cellular membranes with the phosphatidylcholine (PC) reaching up to 55% of total lipids in mammalian cell membranes. Besides glycerophospholipids, another structural lipid class is represented by sphingomyelin (SM). Some glycerophospholipids like phosphatidylinositol (PI), phosphatidylserine (PS), lyso-phosphatidylcholine (LPC) and lyso-phosphatidylethanolamine (LPE) as well as ceramides (Cer) also play key roles in signal transduction pathways [16,17].

Interestingly, a remodeling of the lipidome in PBMCs involving an upregulation of sphingolipids was recently reported in children who later progressed to type 1 diabetes or seroconverted to at least one antibody compared with the autoantibody-negative control children [18]. Lipidomic studies carried out on the plasma of obese patients have shown modulations of some lipid species or classes that have been positively or negatively associated with body mass index (BMI) and/or insulin resistance and glucose intolerance [19]. Similar associations have been observed in prediabetes and T2D [20] where several individual molecular species and lipid classes have been prospectively associated with a risk of T2D [21]. However, very little is known about the modulation of the PBMCs lipidome in these patients.

The main objective of this work was to investigate the impact of obesity with or without dysglycemia on the phospholipidome of PBMCs. Lipidomic analyses focused on glycerophospholipids (PC, PE, PI, PS, LPC, LPE, LPI, LPS), sphingomyelin (SM) and ceramides (Cer) from PBMCs and plasma samples of lean, obese with normoglycemia (OBNG) and obese with dysglycemia (OBDysG) individuals. This lipidomic study highlights for the first-time a disruption of the PBMCs phospholipidome in obese patients with dysglycemia compared to normoglycemic lean or obese individuals.

## 2. Materials and Methods

### 2.1. Participants

We studied a total of 57 individuals (aged 20–63 years) undergoing bariatric surgery or recruited on a voluntary basis. The study was performed in accordance with the ethical principles set forth in the Declaration of Helsinki and received approval from the ethics Committee of the Liège University Hospital, and all patients provided written informed consent (Ethical committee number: 2016/244). Inclusion criteria were based on age (subjects aged 18 to 65), BMI, fasting glucose and glycated hemoglobin (HbA1c) as described below. Subjects with inflammatory or malignant diseases were excluded. Subjects with dementia, mental retardation, altered state of consciousness, not allowing to give their consent were also excluded. Participants underwent measurements of anthropometric parameters and blood pressure (BP) during a medical consultation the day of PBMCs sampling. Body mass index (BMI, kg/m^2^) was calculated from height and weight measurement. Blood pressure was measured using tensiometer. The degree of insulin sensitivity was estimated using the Homeostatic model assessment of insulin resistance (HOMA-IR; fasting plasma glucose [mg/dL] × fasting plasma insulin [mU/L]/405). All biochemical parameters measurements (fasting plasma glucose, lipid/cholesterol, insulin and HbA1c) were performed at Liège university hospital center (CHU Liège) according to ISO 15,189 standards. HbA1c was measured by HPLC on Menarini type HA 8180. The participants were categorized in three groups based on the BMI and glycemic status: (i) lean with normoglycemia (LEAN; BMI < 25 kg/m^2^, fasting glucose < 100 mg/dL and HbA1c < 5.7%; *n* = 14), (ii) obese with normoglycemia (OBNG, BMI ≥ 30 kg/m^2^, fasting glucose < 100 mg/dL and HbA1c < 5.7%; *n* = 18) and (iii) obese with dysglycemia (OBDysG, BMI ≥ 30 kg/m^2^ and fasting glucose ≥ 100 mg/dL and/or HbA1c ≥ 5.7%; *n* = 25), the latter including 18 with prediabetes and 8 with T2D (fasting glucose ≥ 126 mg/dl and/or HbA1c ≥ 6.5%) as defined by the ADA criteria [22]. Of the 25 OBDysG patients, 18 were on glucose-lowering treatment (metformin) and 9 on lipid-lowering agent (statins). No OBNG patient was on glucose-lowering treatment nor on lipid-lowering agent. The participants characteristics are shown in Table 1 and Appendix A.

### 2.2. Sample Preparation

Fasting blood samples were collected in EDTA tubes and centrifuged to collect plasma. Plasma was directly frozen (−80 °C). PBMCs were isolated from the remaining blood using Ficoll-Paque™ PREMIUM (VWR) and immediately frozen at −80 °C.

### 2.3. Lipid Extraction

1 × 10^6^ of PBMCs were sonicated for 10 s in 800 μL water and 100 μL of plasma was diluted with water to reach 800 μL. Then 700 μL of these homogenized samples were then mixed with 800 μL 1N HCl:CH_3_OH 1:8 (*v*/*v*), 900 μL CHCl_3_ and 200 μg/mL of the antioxidant 2,6-di-tert-butyl-4-methylphenol (BHT; Sigma Aldrich). The organic fraction was evaporated using a Savant Speedvac spd111v (Thermo Fisher Scientific, Waltham, MA, USA) at room temperature and the remaining lipid pellet was stored at −20 °C under argon.

### 2.4. Mass Spectrometry

Just before mass spectrometry analysis, lipid pellets were reconstituted in running solution (CH_3_OH:CHCl_3_:NH_4_OH; 90:10:1.25; *v*/*v*/*v*). Phospholipid (PL) species were analyzed by electrospray ionization tandem mass spectrometry (ESI-MS/MS) on a hybrid triple quadrupole/linear ion trap mass spectrometer (4000 QTRAP system; Applied Biosystems SCIEX) equipped with a TriVersa NanoMate (Advion Biosciences) robotic nanosource for automated sample injection and spraying. PL profiling was executed by (positive or negative) precursor ion or neutral loss scanning at a collision energy of 50 eV/45 eV, 35 eV, −35 eV, and −60 eV for precursor 184 (sphingomyelin (SM)/phosphatidylcholine (PC)), neutral loss 141 (phosphatidylethanolamine (PE)), neutral loss 87 (phosphatidylserine (PS)) and precursor 241 (phosphatidylinositol (PI)), respectively. PL quantification was performed by multiple reactions monitoring (MRM), the transitions being based on the neutral losses or the typical product ions as described above. Lipid standards PC25:0, PC43:6, SM30:1, PE25:0, PE43:6, PI25:0, PI31:1, PI43:6, PS25:0, PS31:1, PS37:4, CER35:1, Lyso PC13:0, Lyso PC17:1, Lyso PE13:0, Lyso PE17:1, Lyso PS13:0, Lyso PS17:1, Lyso PI13:0, Lyso PI17:1 (Avanti Polar Lipids) were added based on the amount of DNA in the original sample. Typically, a 3 min period of signal averaging was used for each spectrum. The data were corrected for isotope effects as described by [23].

### 2.5. Lipidomic Statistical Analysis

All analyses were performed using R (version 4.0.5). Individual lipid species were analyzed using principal component analysis with the “prcomp” function. Sums of lipid abundances for each lipid class were calculated and compared across groups (see Appendix A). Normality was checked using (i) Shapiro–Wilk test, (ii) histogram, (iii) QQ-plot. If normality was not respected for at least one of these three tests, normality was rejected. Variance homogeneity was then checked with Levene test. If *p* < 0.05 for Levene test, normality was rejected. Statistical comparisons for biological parameters and lipids abundance (absolute, relative, sums of lipid abundances and ratio) were performed using Kruskal–Wallis non-parametric analysis of variance (ANOVA), followed by Dunn multiple comparison tests due to the abnormal distribution of continuous variables. All data are Mean ± SD and statistical significance was considered when *p* ≤ 0.05.

The effect of different factors such as age, gender, biological variables and their interactions on the lipidome was evaluated. Values were log-transformed prior to linear regression statistical analysis. Associations between biological variables of patient and lipid species were evaluated using linear models (“lm” function), before and after adjusting for the other covariates (age or other biological variables). All *p*-values were adjusted for multiple comparisons using the Benjamini-Hochberg (false discovery rate) method with the “*p*.adjust” function. Relationships were considered significant when the adjusted-*p*-value was ≤ 0.05.

Heatmaps and histograms were created using GraphPad Prism 7. For all analyses, *** *p* ≤ 0.001; ** *p* ≤ 0.01; * *p* ≤ 0.05.

## 3. Results

### 3.1. Anthropometric, Clinical and Biological Characteristics of the Study Population

Anthropometric, clinical and biological characteristics of the study population are summarized in Table 1. Similar to the BMI, the body weight and waist circumference of both OBNG and OBDysG patients are significantly higher than in lean patients. Both obesity phenotypes are characterized by a significant increase in their fasting insulin and HOMA-IR compared to lean patients. However, OBNG and OBDysG patients are distinguished from each other by their fasting glucose and mainly by HbA1c. Indeed, these two metabolic variables are significantly increased in OBDysG patients relative to both OBNG and lean individuals. Both OBNG and OBDysG patients show higher triglyceride and LDL cholesterol levels and lower HDL cholesterol concentrations relative to the lean patients. The pro-inflammatory marker, C-reactive protein (CRP), is also increased in both obesity phenotypes compared to lean individuals. Systolic and diastolic blood pressure (BP), are significantly increased in OBDysG patients relative to lean patients.

Platelet and leukocyte compositions of patients’ blood are summarized in Appendix A. Amounts of blood leukocytes are significantly increased in both groups of obese patients compared to lean patients. Relative values of different leukocyte types do not change depending on obesity. Nevertheless, their absolute amount can vary depending on the group. It is the case for neutrophils and eosinophils which are increased for OBDysG compared to lean patients. Lymphocytes are increased in OBNG patients compared to lean patients.

### 3.2. The Global PBMCs Phospholipidome Is Inversely Associated with Insulin Resistance in Obesity

Lipidomic analyses were performed on plasma and PBMCs from 14 lean individuals, 18 OBNG and, respectively, 24 or 25 OBDysG patients. The global phospholipidome (PC, PE, PS, PI, LPC, LPE, LPS, LPI and SM) as well as ceramides were investigated. A total of 232 and 224 lipid species were detected in PBMCs and plasma, respectively.

As shown in Figure 1A, principal component analysis (PCA) analysis performed on plasma lipidomic data enables to distinguish obese patients from lean individuals while the two obesity phenotypes overlap. Interestingly, the phospholipidome of PBMCs allows to cluster OBDysG and lean patients into two distinct sets with only a small overlap while OBNG individuals are scattered among the other 2 subgroups (Figure 1B).

Then, we have compared the normalized absolute levels (nmol/mL for plasma and nmol/mg DNA for PBMCs) for all detected lipid species in plasma or in PBMCs between each patient’s category. The plasma phospholipidome undergoes important changes in obese versus lean patients; 75 and 54 out of a total of 224 lipid species are significantly up- or down-regulated in lean patients compared to OBDysG and OBNG, respectively (Figure 1C). However, the overall profile is similar in both obese phenotypes; only one lipid species (PI 36:4) is significantly differentially regulated between both obese groups (Figure 1C). On the other hand, unlike OBNG individuals, the global PBMCs phospholipidome from OBDysG patients is significantly downmodulated compared to lean patients (Figure 1D, Appendix A); 163 out of a total of 232 lipid species are significantly reduced in OBDysG compared to lean patients against only 10 when OBNG and lean individuals are compared.

The whole plasma lipidome, unlike that of PBMCs, has already been the subject of much work in the context of obesity and prediabetes/T2D. Therefore, we decided not to further investigate the plasma phospholipidome and to focus on the modulation of PBMCs phospholipidome in the rest of this work.

The down-modulation of the PBMCs phospholipidome in OBDysG patients affects all classes of phospholipids. The PC, PE, LPC, LPE and SM classes are the most impacted; almost all species of PC (50 out of 55, i.e., 91%, Figure 2A), PE (30 out of 42, i.e., 71%, Figure 2B), LPC (13 out of 19, i.e., 68%, Figure 2C), LPE (13 out of 13, i.e., 100%, Figure 2C) and SM (16 out of 21, i.e., 76%, Appendix A) are significantly down-regulated in OBDysG compared to lean individuals. In the case of PE and PI, the down-modulation targets mostly the species carrying more than 2 or 3 double bonds, respectively (Figure 2B, Appendix A). A few PS species are also affected without showing a clear pattern with respect to the acyl chain composition (Appendix A) while less than half of LPI and LPS species are affected (Appendix A). In contrast, ceramides seem to respond differently than phospholipids; indeed, some species are down-modulated only in OBDysG patients while others decrease significantly in both types of obese patients (Appendix A).

Except for both LPS and LPI classes, the absolute abundance of each lipid class is significantly downregulated in OBDysG in comparison to lean patients while there is no significant difference when OBNG are compared to lean individuals (Figure 3). Nevertheless, regarding LPE and LPI classes, a significant downregulation is observed in OBDysG compared to OBNG patients. Despite this general decrease in the absolute abundance of each lipid class, the relative proportions of each lipid class are preserved, except for the LPE where relative abundance is significantly reduced in OBDysG compared to lean patients (2.2% vs. 3.8%, * *p* ≤ 0.05) (Appendix A).

We wanted to determine whether the changes in the phospholipidome were associated with differences in anthropometric, metabolic and inflammatory parameters. Many plasma lipid species are known to be associated with age, gender and clinical lipids (total cholesterol, HDL-cholesterol and triglyceride) [24]. What about PBMCs phospholipids? In univariable analyses, gender, HDL, LDL and triglycerides are not associated with any PBMCs lipid species or class. However, several phospholipid classes (PC, PE, SM and LPE) are inversely associated with age (Table 2). A similar trend is observed for the Cer, LPC and PI (Table 2). This negative association with age is particularly important for the LPEs. Almost all individual LPEs’ lipids (11 out of 13) are associated with age, justifying the significant decrease in the relative abundance of this lipid class in OBDysG patients which present higher mean age compared to the two other subgroups. In univariate analyses, the main phospholipids classes are also negatively associated with BMI, CRP, HOMA-IR, fasting insulin and HbA1c (Appendix A and Appendix A). The relationships with BMI and HbA1c are lost after adjustment for age (Appendix A and Appendix A) while they are maintained in the case of CRP (Appendix A). Interestingly, strong negative associations between the PC, PE, PI, Cer and SM are observed with both parameters allowing to assess insulin resistance, namely HOMA-IR and fasting insulin. Moreover, these relationships are retained after correcting for age in multivariate analyses (Appendix A). Any association with fasting glucose is observed (Appendix A). Altogether, these multiple linear regression analyses demonstrate that the global PBMCs phospholipidome in obesity is not associated with dysglycemia estimated by fasting glucose and HbA1c but rather with insulin resistance as demonstrated by a strong negative association with HOMA-IR and especially with fasting insulin. Accordingly, we analyzed the associations of all individual lipid species from each class with fasting insulin after age adjustment. The β-coefficients and 95% confidence intervals are converted to percentage change for interpretation of results (Figure 4). The lipid species which are the most influenced by fasting insulin levels (negative correlation; *p* < 0.05 or <0.01) are PC (39 out of 55 species) and PE (23 out of 42 species) (Figure 4). Interestingly, the PC and PE lipid species with less than 40 carbon atoms are most strongly associated with insulin (*p* < 0.01). The levels of some of these PC and PE species decrease by more than 40% as the insulin concentration increases by 1 mU/L (Figure 4). Moreover, no lipid species are positively associated with fasting insulin (Figure 4).

### 3.3. A Few Individual Phospholipidic Species Allow to Distinguish Obese Patients with Dysglycemia from Normoglycemic Obese Individuals

Sixteen lipid species belonging to the PC (28:0), SM (d18:1/22:0, d18:1/20:1), LPC (20:0, 20:3), LPE (18:1, 20:1, 16:0, 18:2, 20:2, 20:3, 22:3, 20:4) and LPI classes (18:0, 20:4) are significantly downmodulated in OBDysG compared to OBNG patients, making possible to distinguish the 2 obese phenotypes (Figure 5).

Univariate and multivariate linear regression analyses were also performed with these lipid species. As previously shown for the LPE class, the 8 LPE species differentially modulated between OBDysG and OBNG patients are inversely associated with age (Table 2). These species do not show any correlation with any patient characteristics after age adjustment except the LPE 16:0 that remains inversely associated with HOMA-IR and fasting insulin (Appendix A). However, the other lipid species, namely the PC 28:0, both LPC (20:0, 20:3), both SM (d18:1/20:1, d18:1/22:0) and both LPI (18:0, 20:4), are not correlated with age (Table 2). In univariate linear regression analyses, all of these lipid species, with the exception of the LPI 20:4, are negatively associated with insulin resistance (HOMA-IR, fasting insulin) and, for the most part, also with BMI and CRP (Appendix A). In addition, LPC 20:0 and both SM (d18:1/20:1, d18:1/22:0) are inversely associated with HbA1c and the PC 28:0 is negatively correlated with both fasting glucose and HbA1c (Appendix A).

### 3.4. The Relative Abundance of Saturated PC Is Increased and the Lyso-PC/PC Ratio Is Decreased in PBMCs from OBDysG Versus OBNG Patients

We also investigated whether the OBNG and/or OBDysG phenotypes are associated with changes in the composition of the PL acyl chains. The proportions of PL classified according to their number of carbon atoms are not affected by obesity nor by dysglycemia (Figure 6A). However, the percentage of PL with 0 unsaturation (i.e., saturated) is significantly increased in OBDysG compared to OBNG while the frequency of PL with 2 unsaturations is decreased in OBDysG versus lean patients (Figure 6B). This up-regulation of the relative abundance of saturated PL is mainly due to the increase in the percentage of saturated PC even if this signature is also visible for PE (Figure 6C). The PI are not represented as no saturated PI was detected.

The percentage of saturated PE is only positively associated with age (unadjusted β = 0.359; 95% CI (0.13, 0.59); *p* = 0.046). However, the percentage of saturated PC is not correlated neither with age, BMI, CRP nor with HOMA-IR and fasting insulin but shows a positive association with HbA1c (Figure 6D).

Because previous studies suggest that altering the LPC/PC ratio of cell membranes is sufficient to disrupt the physical properties of membranes [25], we also focused on the Lyso-PL/PL ratios. Interestingly, the LPC/PC ratio is significantly decreased in OBDysG individuals when compared to OBNG patients (Figure 7). A significant decrease in the LPE/PE ratio is also observed in OBDysG compared to OBNG individuals although the inter-individual variability in each patient category is important (Figure 7). While the ratio LPE/PE is only negatively associated with age, the LPC/PC ratio is not influenced by any of the previously studied clinical variables.

## 4. Discussion

In this study, we identified for the first time a distinct phospholipidome in PBMCs from obese patients with dysglycemia compared to normoglycemic obese or lean individuals. We demonstrated a downmodulation of the global phospholipidome in PBMCs from OBDysG compared to lean patients which is mainly associated with insulin resistance (i.e., HOMA-IR and fasting insulin). The main PL classes constituting the lipid bilayer of mammalian cell membranes, namely the PC, SM and PE, are strongly downmodulated suggesting a shrinkage of the membrane network. Their relative abundance seems to be preserved, notably the PE/PC ratio (data not shown) playing a key role in cell membrane homeostasis [26,27]. However, we observed that the LPC/PC ratio is significantly decreased in PBMCs from OBDysG compared to OBNG patients. Interestingly, this LPC/PC ratio has just been proposed as a factor capable of modulating the physical properties of cell membranes and the activity of membrane receptors. Indeed, a lipidomic analysis recently performed on primary myocytes from individuals that are insulin-sensitive and lean or insulin-resistant with obesity (OB) also revealed a global phospholipidome down-modulation in OB myotubes [25]. Authors underlined the specific decrease in the LPC/PC ratio in these OB myotubes that resulted from a greater expression of lysophosphatidylcholine acyltransferase 3 (LPCAT3), an enzyme involved in phospholipid transacylation (Lands cycle). This disruption of the LPC/PC ratio induced a disorganization of membrane phospholipids that was able to suppress insulin receptor phosphorylation [25].

Furthermore, we observed an increased percentage of saturated PC in the membranes of PBMCs from OBDysG compared to OBNG patients. An upregulation of the phospholipid acyl-chain saturation index is well-known to impair the membrane fluidity [26,28]. A decreased membrane fluidity has been already described in red blood cells and leucocytes in patients with T2D [29]. Such lipid bilayer stress can influence the activity of integral proteins. In this context, we previously demonstrated that incorporation of saturated fatty acids into PC induces a disruption of the plasma membrane Na/K-ATPase followed by K+ efflux and NLRP3 inflammasome-mediated IL-1β secretion in human macrophages [28].

A few lipid species are significantly down-modulated in the PBMCs of OBDysG compared to OBNG individuals allowing us to discriminate these two categories of patient; this is the case for the PC 28:0, the LPC 20:0 and both SM d18:1/20:1 and d18:1/22:0. Moreover, these species are inversely correlated to obesity (BMI), insulin resistance (HOMA-IR, fasting insulin) and also dysglycemia (HbA1c). Thus, these PBMCs lipid species could be potential novel predictors of glucose intolerance in obesity.

Altogether, these data suggest a shrinkage of the membrane network and membrane physico-chemical alterations. Thus, such phospholipidome remodeling could disrupt the function of circulating immune cells. Indeed, immune cells need to expand their ER and Golgi membranes when they have to synthesize and secrete large amounts of immunoglobulin or cytokines. T lymphocytes and monocytes are activated in response to the stimulation of membrane receptors like T-cell receptor (TCR) or Toll-like receptor (TLR), respectively; both kinds of receptors have been reported to be disturbed by changes in the biophysical properties of membranes [26,30]. These data raise many questions. We do not know if all cell membranes are impacted in PBMCs, membranes of organelles such as ER and Golgi and/or the plasma membrane and whether all cell subpopulations, lymphocytes and/or monocytes, are affected. Further lipidomic analyses performed on various cellular fractions and on the different PBMCs subpopulations will allow to answer these questions.

What could be the triggering events of such a phospholipidome remodeling? Given the global phospholipidome decrease, we can suspect that the glycerophospholipid synthesis pathways are downmodulated in the PBMCs of OBDysG patients. The SM are also globally affected and related with PC and PE through the same biosynthetic pathway. Thus, we can assume that the decrease in SM levels is due to a lack of precursors. Because the Lyso-PL/PL ratios are also altered, the Lands cycle enzymes could also be differentially modulated in the PBMCs of OBDysG patients, as suggested by Ferrara et al. [25]. An increasingly defended hypothesis suggests that systemic metabolism disturbances in obese patients with insulin resistance or glucose intolerance can impact the intrinsic metabolism of PBMCs through altered fuel supplies [10,11]. Modulations of some lipid classes have already been observed in circulating immune cells of obese patients; this is the case of TG which accumulate in lipid droplets of NK cells during obesity causing complete ‘paralysis’ of their glucose metabolism and cytotoxic activity [12]. Interestingly, FFAs added to the culture medium of healthy NK cells are able to recapitulate the activation of the TG synthesis pathways and the NK cells ‘paralysis’ [12]. FFAs are released in the blood of obese patients as a result of an exacerbated lipolysis in adipocytes due to insulin resistance. Therefore, FFAs concentrations are higher in patients with prediabetes or T2D than in normoglycemic obese [31]. Since we report that the majority of glycerophospholipids and SM species of PBMCs are strongly negatively associated with insulin, we cannot exclude a role of blood FFAs or a direct effect of insulin on intrinsic lipid metabolism of leukocytes from OBDysG patients. Transcriptomic analyzes coupled with the investigation of the whole lipidome on PBMCs from OBDysG, OBNG and lean patients as well as on healthy PBMCs treated ex vivo with FFAs or insulin will make possible to answer these questions.

The molecular mechanisms underlying the increase in PC saturation in OBDysG patients are still elusive. We show in this work a positive correlation between the percentage of saturated PC and HbA1c. A hyperglycemic environment could exacerbate the glucose metabolism and its conversion into saturated fatty acids via de novo lipogenesis in leucocytes. It would be interesting to check the association between the percentage of saturated PC and HbA1c on a larger cohort comprising not only obese patients with and without dysglycemia but also lean individuals with or without glycemia disorders.

### Limitations

This study has some limitations. Indeed, this cohort of patients was initially recruited to perform other analyzes on PBMCs. Hence, the study is not powered for lipidomic analyses and our sample size may have been too small to detect significant correlations for some variables. Further lipidomic analyses should be performed on a larger cohort whose subgroups are better age- and sex-matched.

The majority of OBDysG patients in our study (18 of 25) were on treatment with metformin, unlike OBNG and lean patients. Metformin is the first-line drug for T2D treatment. However, its plasma glucose level-lowering effects are still not fully understood. Metformin would mediate its primary antidiabetic action by preventing hepatic glucose production through inhibition of the mitochondrial respiratory chain complex 1 [32]. The activation of AMP-activated protein kinase (AMPK) by metformin was reported to inhibit lipid synthesis in hepatocytes and mitigate hepatic steatosis and insulin resistance in fatty liver mouse models [33]. However, it is not known whether this mechanism is relevant in humans. Until now, nobody reported an impact of metformin on the lipid metabolism in PBMCs from treated patients. However, to rule out this hypothesis, we compared the levels of each phospholipid species or class as well as the percentage of saturated PC in the PBMCs of metformin-treated (*n* = 18) and -untreated (*n* = 7) OBDysG patients. No difference could be observed in the PBMCs phospholipidome of these two groups of patients (data not shown).

Of the 25 OBDysG patients, 9 were on lipid-lowering agent (statins) treatment. Again, when we compared the levels of all PBMCs phospholipid species, we did not observe any difference between OBDysG patients with or without statin treatment (data not shown).

## 5. Conclusions

Notwithstanding these limitations, we report for the first time a disruption of the PBMCs phospholipidome in obese patients with dysglycemia compared to normoglycemic lean or obese individuals. A global decrease in the PBMCs phospholipidome is observed in OBDysG patients and is strongly associated with insulin resistance. Some individual lipid species that are downregulated in OBDysG compared to OBNG individuals seem to be further associated with dysglycemia and could be potential novel predictors of glucose intolerance in obesity. Interestingly, an enrichment of saturated PC is observed in PBMCs from OBDysG patients compared to OBNG and is positively associated with HbA1c. The PBMCs phospholipidome remodeling in OBDysG patients involving both an increased saturated PC percentage and a decreased LPC/PC ratio could disrupt membrane homeostasis and immune cell function. Further lipidomic analyzes should be extended to all lipid classes (including TG, glycosphingolipids) and performed on leukocyte subpopulations. Using an integrative approach with functional, lipidomic and transcriptomic analyses would make possible to elucidate the origins of lipidome remodeling and its consequences on immune cells.

## Figures and Tables

**Figure 1 nutrients-13-03461-f001:**
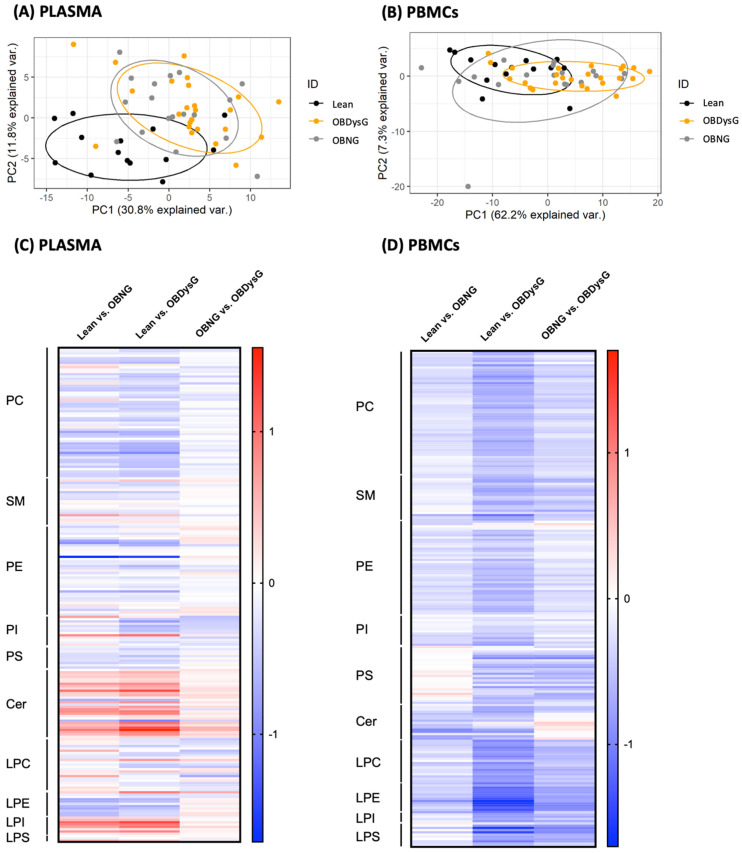
Comparison of both plasma and peripheral blood mononuclear cells (PBMCs) phospholipidome between lean, OBNG and OBDysG patients. Principal component analysis score plot of the lipidomic data (PC, PE, PI, PS, SM) obtained on (**A**) the plasma or (**B**) PBMCs from lean (*n* = 14), OBNG (*n* = 18) and OBDysG (plasma *n* = 24 and PBMCs *n* = 25) patients. Log2FC of all lipid species detected in (**C**) the plasma and (**D**) PBMCs between lean and OBNG or OBDysG patients and between the two obesity phenotypes (OBNG vs. OBDysG).

**Figure 2 nutrients-13-03461-f002:**
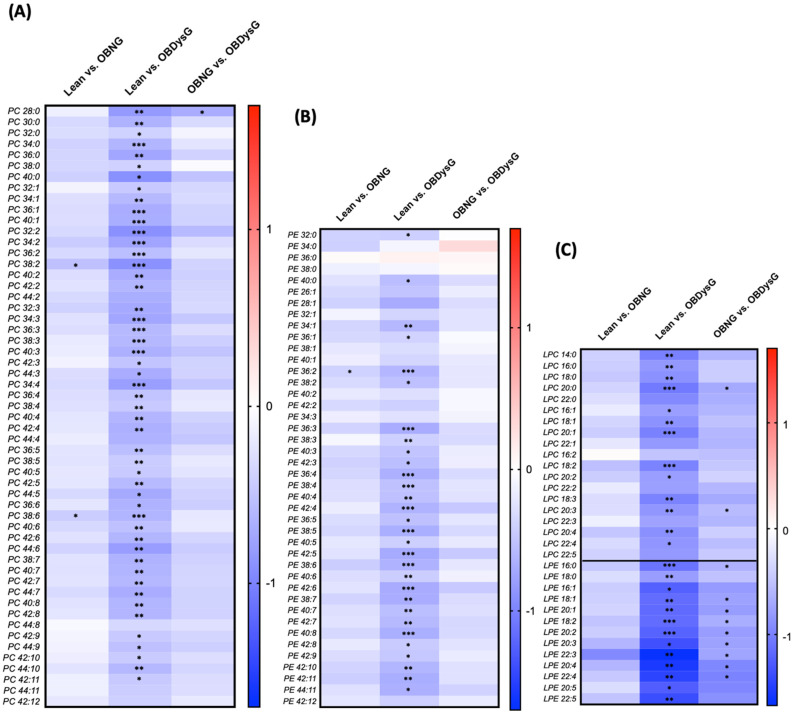
Log2FC of all (**A**) PC, (**B**) PE and (**C**) Lyso-PC/-PE lipid species detected in PBMCs between lean and OBNG or OBDysG patients and between the two obesity phenotypes (OBNG vs. OBDysG). Mean comparison * *p*≤ 0.05; ** *p* ≤ 0.01; *** *p* ≤ 0.001. Kruskal-Wallis followed by Dunn post hoc test was performed on data.

**Figure 3 nutrients-13-03461-f003:**
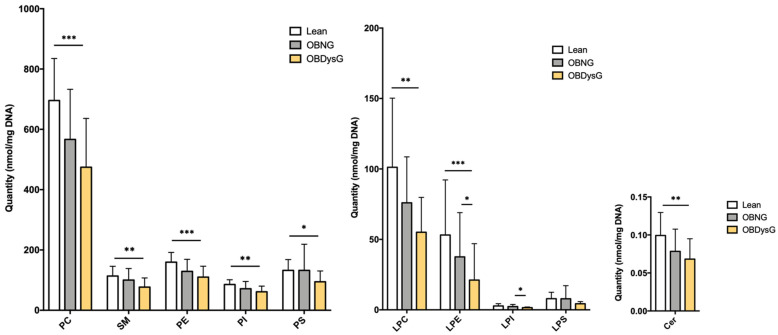
Absolute abundance (nmol/mg DNA) of each lipid class in PBMCs from lean, OBNG and OBDysG patients. Data are mean ± SD. * *p* ≤ 0.05; ** *p* ≤ 0.01; *** *p* ≤ 0.001. Kruskal–Wallis followed by Dunn post hoc test was performed on data. Lean (*n* = 14), OBNG (*n* = 18) and OBDysG (*n* = 25).

**Figure 4 nutrients-13-03461-f004:**
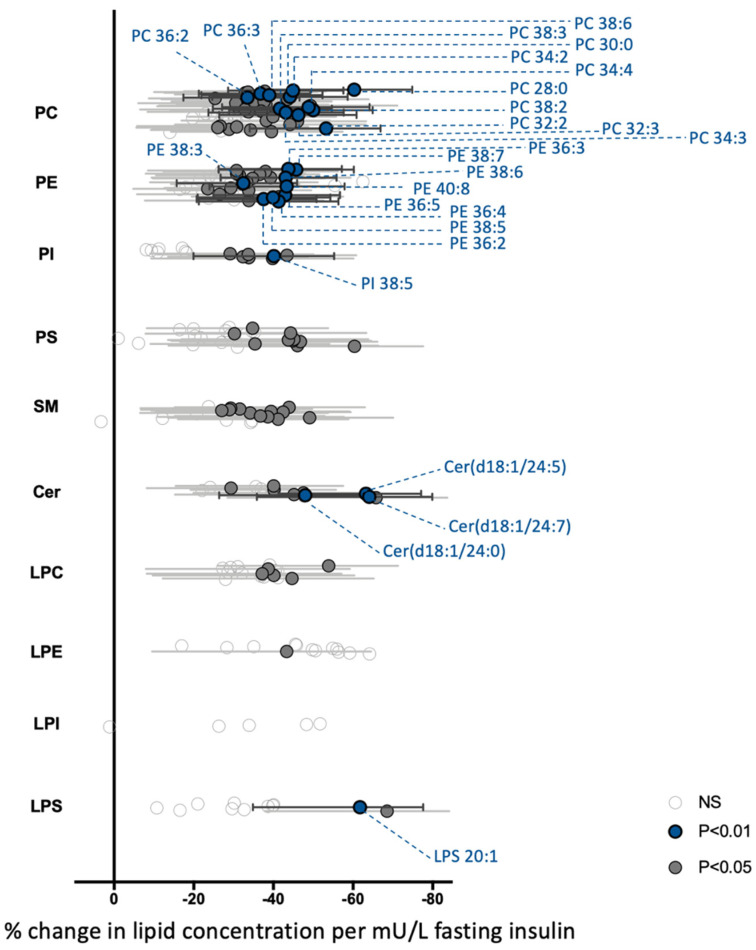
Association between fasting insulin and all detected PBMCs lipid species. Linear regression analysis between fasting insulin and each lipid species was performed on 14 lean, 18 OBNG and 25 OBDysG individuals adjusting for age. Gray open circles show lipid species with non-significant association with fasting insulin, dark gray and blue circles show species with significant association, *p* < 0.05 and *p* < 0.01, respectively, after correction for multiple comparisons (Benjamini-Hochberg). β-coefficients and 95% confidence intervals were then converted to percentage change (percentage change = (10^ β-coefficient−1) × 100) for interpretation of results.

**Figure 5 nutrients-13-03461-f005:**
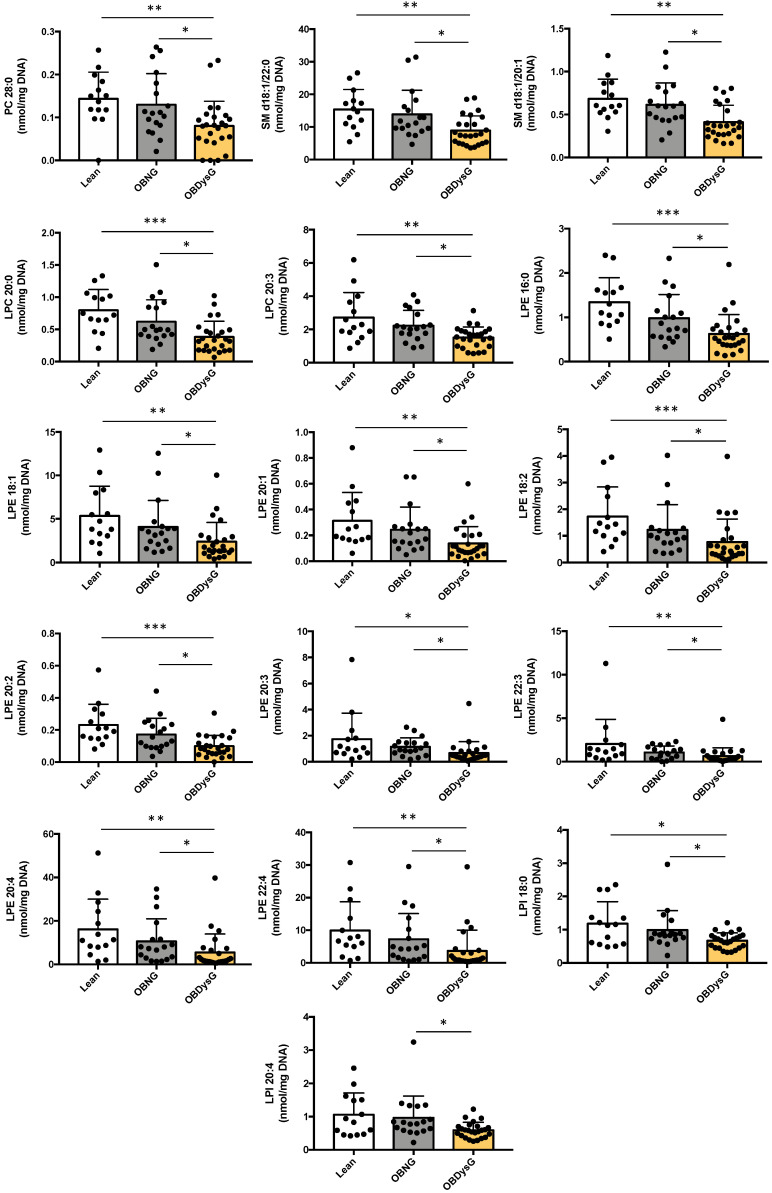
Individual lipid species downmodulated in PBMCs from OBDysG compared to OBNG patients. Data are mean ± SD. * *p* ≤ 0.05; ** *p* ≤ 0.01; *** *p* ≤ 0.001. Kruskal–Wallis followed by Dunn post hoc test was performed on data.

**Figure 6 nutrients-13-03461-f006:**
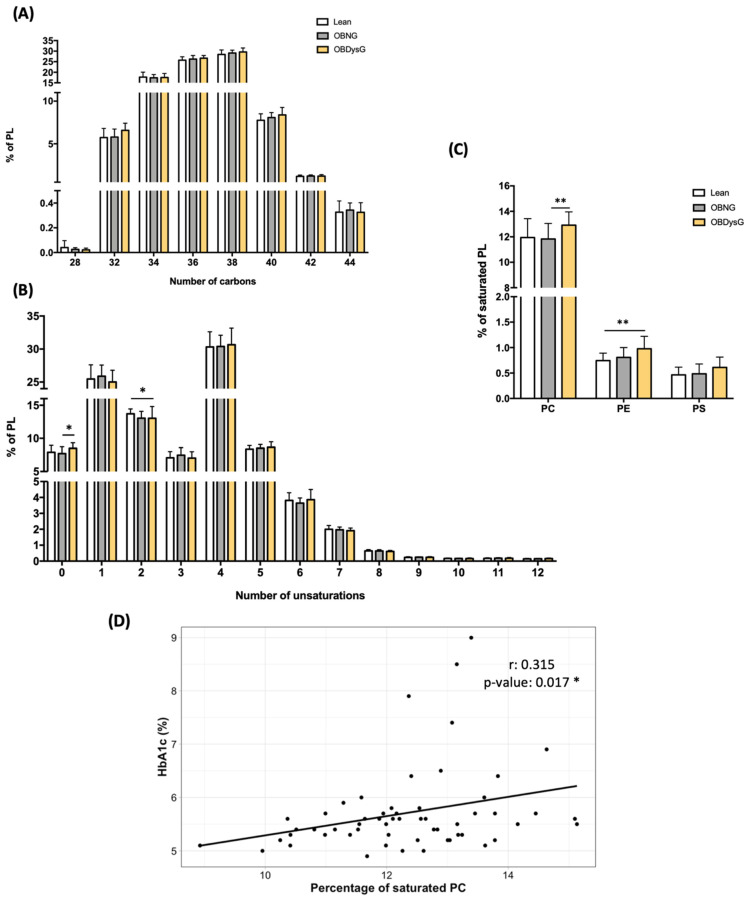
The percentage of saturated PC is increased in PBMCs from OBDysG compared to OBNG patients. Relative abundance of the PL species (PC + PE + PI + PS) comprised of (**A**) the same number of carbons or (**B**) the same number of unsaturation in their hydrocarbon chain and (**C**) relative abundance of saturated PC, PE or PS. Data are mean ± SD. Kruskall–Wallis test, Dunn’s multiple comparison test, * *p* < 0.05; ** *p* < 0.01. (**D**) Correlation between HbA1c (%) and percentage of saturated PC. *n* = 54, r = Spearman’s rank correlation coefficient.

**Figure 7 nutrients-13-03461-f007:**
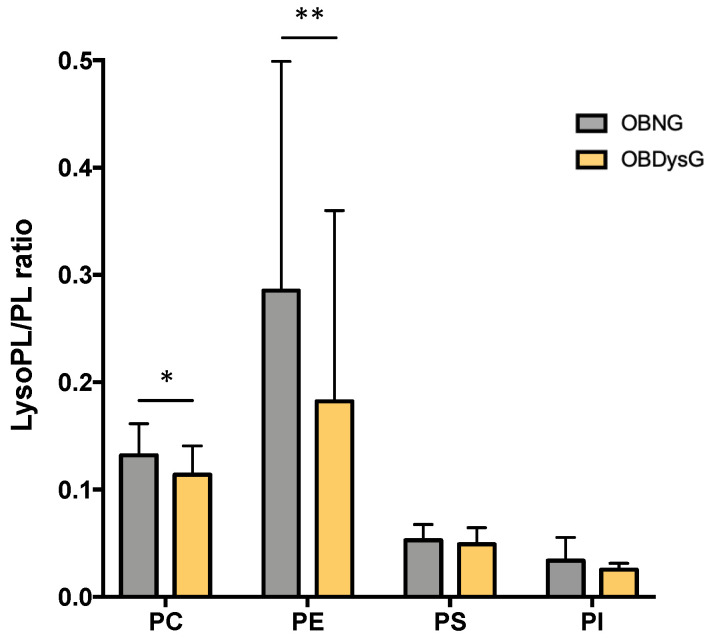
The Lyso-PC/PC and Lyso-PE/PE ratios are decreased in PBMCs from OBDysG compared to OBNG patients. Data are mean ± SD. Mann–Whitney-Wilcoxon Test was performed on data. OBNG (*n* = 18) and OBDysG (*n* = 25). * *p* < 0.05; ** *p* < 0.01.

**Table 1 nutrients-13-03461-t001:** Anthropometric, clinical and biological characteristics of the participants.

	Overall *p*-Value	Lean	OBNG	OBDysG
*n* (f/m)	NA	14 (9/5)	18 (12/6)	25 (12/13)
Age (years)	>0.001	33.9 ± 9.7	39.1 ± 8.9	47.9 ± 11.3 ***^;^†
Body Weight (kg)	>0.001	65.5 ± 10.5	112.7 ± 22.1 ***	120.6 ± 25.4 ***
BMI (kg/m^2^)	>0.001	22.1 ± 1.9	40.3 ± 4.4 ***	40.6 ± 6.6 ***
Waist (cm)	>0.001	76.2 ± 8.1	121.0 ± 13.9 ***	124.5 ± 14.9 ***
Systolic BP (mmHg)	0.003	122.3 ± 9.2	129.8 ± 13.9	140.9 ± 17.3 **
Diastolic BP (mmHg)	0.024	74.0 ± 9.2	83.2 ± 10.4	87.6 ± 16.1 *
Fasting glucose (mg/dL)	>0.001	84.6 ± 5.3	90.2 ± 5.4	112.0 ± 32.5 ***^;^†
Fasting insulin (mU/L)	>0.001	7.1 ± 3.3	22.9 ± 10.7 ***	32.6 ± 22.2 ***
HOMA-IR	>0.001	1.48 ± 0.68	5.14 ± 2.43 ***	9.67 ± 8.79 ***
HbA1c (%)	>0.001	5.27 ± 0.23	5.37 ± 0.20	6.20 ± 1.00 ***^;^†††
Type 2 diabetes	NA	0/14	0/18	8/25
Triglycerides (g/L)	>0.001	0.81 ± 0.35	1.49 ± 0.50 **	1.87 ± 1.77 ***
Total cholesterol (g/L)	0.570	1.94 ± 0.31	2.05 ± 0.30	2.05 ± 0.41
HDL cholesterol (g/L)	>0.001	0.76 ± 0.16	0.49 ± 0.10 ***	0.45 ± 0.10 ***
LDL cholesterol (g/L)	0.015	1.02 ± 0.31	1.29 ± 0.30 *	1.29 ± 0.35 *
C-reactive protein (mg/L)	0.001	2.12 ± 2.09	6.17 ± 5.88 *	7.33 ± 6.05 ***

Data are mean ± SD. Kruskal–Wallis followed by Dunn post hoc test was performed on data. Normoglycemic obese (OBNG) or obese with dysglycemia (OBDysG) vs. Lean * *p*≤ 0.05; ** *p* ≤ 0.01; *** *p* ≤ 0.001. OBDysG vs. OBNG † *p*≤ 0.05; †† *p* ≤ 0.01; ††† *p* ≤ 0.001.

**Table 2 nutrients-13-03461-t002:** Univariable associations between selected lipid species or classes and age.

Age			
Lipid class	Unadjusted β (95% CI)	*p*	R^2^ (*p*-value)
PC	−0.411 (−0.70, −0.13)	0.051	0.111 (0.007)
PE	−0.386 (−0.67, −0.10)	0.051	0.097 (0.010)
PI	−0.350 (−0.64, −0.06)	0.059	0.078 (0.020)
PS	−0.246 (−0.64, 0.15)	0.291	0.008 (0.230)
SM	−0.467 (−0.81, −0.12)	0.051	0.098 (0.010)
Cer	−0.367 (−0.71, −0.03)	0.081	0.058 (0.040)
LPC	−0.534 (−0.97, −0.09)	0.059	0.077 (0.021)
LPE	−1.238 (−2.00, −0.47)	0.046	0.139 (0.002)
LPI	−0.320 (−0.78, 0.14)	0.238	0.015 (0.177)
LPS	−0.168 (−0.69, 0.36)	0.581	−0.011 (0.531)
**Lipid species**	**Unadjusted β (95% CI)**	** *p* **	**R^2^ (*p*-value)**
PC 28:0	−0.445 (0.11, −1.00)	0.183	0.028 (0.122)
SM d18:1/20:1	−0.445 (−0.86, −0.03)	0.082	0.056 (0.042)
SM d18:1/22:0	−0.548 (−1.00, −0.09)	0.059	0.075 (0.022)
LPC 20:0	−0.623 (−0.05, −1.19)	0.076	0.060 (0.036)
LPC 20:3	−0.410 (0.05, −0.87)	0.135	0.036 (0.084)
LPE 16:0	−0.913 (−0.38, −1.45)	**0.046**	0.155 (0.001)
LPE 18:1	−1.098 (−0.45, −1.74)	**0.046**	0.154 (0.002)
LPE 20:1	−1.201 (−0.44, −1.96)	**0.046**	0.133 (0.003)
LPE 18:2	−1.199 (−0.49, −1.91)	**0.046**	0.150 (0.002)
LPE 20:2	−1.013 (−0.25, −1.77)	**0.052**	0.094 (0.011)
LPE 20:3	−1.192 (−0.43, −1.96)	**0.046**	0.129 (0.004)
LPE 22:3	−1.329 (−0.45, −2.21)	**0.046**	0.125 (0.005)
LPE 20:4	−1.445 (−0.45, −2.44)	**0.051**	0.111 (0.006)
LPE 22:4	−1.625 (−0.55, −2.71)	**0.046**	0.121 (0.005)
LPI 18:0	−0.264 (0.18, −0.71)	0.309	0.006 (0.252)
LPI 20:4	−0.385 (0.09, −0.86)	0.178	0.027 (0.118)

Data presented as β-coefficients and corresponding 95% confidence intervals (CI). All lipids were log-transformed to the base 10 prior to analyses and all *p*-values represent significance of associations after correcting for multiple comparisons using the Benjamini-Hochberg method (bold values indicate *p* < 0.05 after correction).

## Data Availability

The raw lipidomic data presented in this study are openly available in Zenodo at https://doi.org/10.5281/zenodo.5226515 (accessed on 20 August 2021).

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
