# Peer review of "New Insights on the PBMCs Phospholipidome in Obesity Demonstrate Modulations Associated with Insulin Resistance and Glycemic Status"

_nutrients, 2021, doi:10.3390/nu13103461_

Round 1
Reviewer 1 Report
It has long been known that obesity induces chronic systemic inflammation, in part driven by macrophage infiltration into visceral adipose tissue because of “leaky” adipocytes. Cellular metabolism can become progressively dysregulated, leading to problems not only with glycemic regulation in skeletal muscle and the brain but leukocyte development and mobilization. A key piece of missing evidence is that while free-fatty acids and triglycerides are used as “cheap” but ultimately harmful sources of energy in lieu of adequate glucose utilization, the lipidome of PBMCs in obesity with or without dysglycemia has not been investigated. This is important because dysglycemia once established is progressive, difficult to treat, and has profoundly negative cascading effects on immunity.
The cardiometabolic factors, PBMC isolation, and ESI-MS are all standard or conducted using standard protocols. Isolation of the lipid fraction is one of many variants of the well established Bligh and Dyer method. The targeted MS for various lipid species all looks fine. For the participants themselves, Table 1 indicates that the OBNG group had on average Class III Obesity but a surprisingly low and invariant fasting glucose level.
For the statistical approach, the following is a mild concern. One is always free to use a non-parametric test and not do parametric statistics. Even so, it is not clear why MANOVA/MANCOVA was not used to test for Lipid Class differences in the aggregated lipid species variables themselves, instead of a factor-analysis solution. Violations of normality would reasonably lead to using K-W. MANOVA is helpful because if one’s data assumptions hold, the initial MANOVA test itself—the omnibus test, some call it—can lead to post-hoc testing that does not require multiple correction. Given the sample size and number of outcome measures, I worry that a lack of Power might lead to some interesting results being rejected due to type 2 error. Separately, while PCA likely recapitulated actual lipid classes fairly well, there’s nonetheless some variance lost that would be better accounted for using MANOVA.
The results are fascinating, but I first have some friendly, basic points of curiosity for which I would very much appreciate some insight. Participants in the OBNG and OBDysG groups have, on average, Class III obesity. This obviously makes sense because some participants were patients undergoing bariatric surgery. Even if they’re young to middle-aged, I am astonished that the OBNG group has such low, stable fasting glucose values, particularly since they clearly have insulin resistance. For the clinicians among the co-authors, is this cardiometabolic profile in Belgium seen regularly? Or is it unusual? Many OBDysG participants were on medications for controlling hyperglycemia, but were any OBNG? In the States, when we do clinical work in community members, I see some borderline obese in their late teens or 20’s with normoglycemia but then their insulin values are also low. I may be missing something obvious, but this fascinates me.
It's fair to focus on PBMC differences instead of plasma, which I agree has received a great deal of attention and is not a necessary component for a report focused on the unknown question of lipid profiles in PBMCs when hyperglycemia is or is not present.
What is equally interesting is that lean vs. OBDysG shows so many differences in lipid species expression, but that lean vs. OBNG shows almost none. This is a remarkable set of findings, because it suggests that insulin resistance and therefore hyperinsulinemia in and of itself may have no role to play for whether PBMC initially begin to “shed” lipid species that span a wide variety of cellular features. But that HOMA-IR and therefore insulin resistance itself drives this drastic shift in lipid evacuation once dysglycemia has begun. This makes sense intuitively. While the intracellularly cascade of IRS-1 -> PI3K -> AKT grinds to a halt and therefore insulin-mediated glucose utilization, receptor sensitivity can vary. Excess glucose and excess insulin that shows insulin resistance, but perhaps not intracellular IR, may drive PBMCs to not use lipids. This is similar to the discussion on page 16, although the authors are more elegant in their discussion of the cellular mechanisms than I could be.
Regardless, the data is convincing and I think could provide many interesting avenues of research into how immune cells metabolically “switch.” I am most keenly interested if this happens in the brain, and if this is site-specific based on insulin-insensitive vs. sensitive glucose receptor density.
Reviewer 2 Report
- Please provide the number of the bioethical committee resolution.
- Please describe in detail the inclusion and exclusion criteria.
- Why did the authors include in the study people aged 20-66 years?
- Please provide more details about the recruitment process. When was the study conducted? Who did perform the recruitment and qualification to the study?
- Please provide methods of biochemical parameters measurements.
- Please describe how the anthropometric parameters and blood pressure were measured.
- Did the authors analyze the body composition of the study population?
- Have the authors calculated the minimum sample size or study power?
- How the authors checked the normality of the distribution of variables?
Round 2
Reviewer 2 Report
The authors have successfully addressed most of my concerns.